# Fiber Lateral Pressure Sensor Based on Vernier– Effect Improved Fabry–Perot Interferometer

**DOI:** 10.3390/s22207850

**Published:** 2022-10-16

**Authors:** Xu Guo, Rui Wu, Jingcheng Zhou, Andres Biondi, Lidan Cao, Xingwei Wang

**Affiliations:** 1Department of Electrical and Computer Engineering, University of Massachusetts Lowell, Lowell, MA 01854, USA; 2Department of Nano Engineering, University of California, 9500 Gilman Drive, San Diego, CA 92092, USA

**Keywords:** fiber optic sensor, Vernier effect, Fabry–Perot

## Abstract

A fiber optic pressure sensor that can survive 2200 psi and 140 °C was developed. The sensor’s pressure sensitivity was measured to be 14 times higher than bare FBG when tested inside stacks of ultra-high-molecular-weight polyethylene (UHMWPE) composite fabric. The sensitivity can be further improved 6-fold through the Vernier effect. Its tiny sensing length (hundreds of microns) and uniform outer diameter (125 µm) make it a suitable candidate for real-time point pressure monitoring under harsh environments with limited space, such as in composite-forming procedures.

## 1. Introduction

Composite material components are widely applied in the civil structure engineering, aerospace, transportation and manufacturing industries [1,2,3]. Composite material manufacturing processes usually face external strain and harsh environmental conditions, such as high temperature and pressure. Thus, the real-time monitoring of composite material structural health is needed. Fiber optic sensors are suitable for composite material structural health monitoring given their advantages of compact size and durability [4,5,6,7,8,9,10]. While significant research has been devoted the consistency and efficiency of this process [11,12,13,14], there are still numerous defects that may occur during the manufacturing process, including delamination of the layers. Currently, there is no method to measure the pressure applied during the manufacturing process and the real-time deformation of the composite material. To enable better quality control of the manufacturing process in harsh environments, sensors for measuring high pressure (up to 1900 psi) are required. A suitable sensor must have a small size to prevent interference in the formation of the composite, and be able to survive high temperatures (120 °C) and pressures.

Fiber lateral pressure sensors can measure lateral pressure exerted on fiber. Compared with fiber tip pressure sensors, they have better robustness and larger measurement ranges because they are built inside an optic fiber. On the contrary, their sensitivities are lower than fiber tip pressure sensors due to this structure. Thus, they are typically used in areas where robustness and pressure measuring range are essential. The Fiber Bragg grating (FBG) sensor is a typical example, generally used as a pressure sensor in these fields. It converts the ambient pressure to the deformation of fiber grating structure. However, it suffers from low pressure sensitivity, which is normally 3.04 pm/MPa [15].

Much effort has been devoted to improving such sensors’ sensitivity and survivability. Wu et al. wrote the FBG structure on a grapefruit micro-structured fiber to achieve a sensitivity of 12.86 pm/MPa [16]. However, this FBG sensor was written on specially designed fiber, which was not commercially available and thus limited the sensor’s applications. Photonic crystal fibers (PCF) with novel structures were also used to fabricate lateral pressure sensors. Liu et al. manually spliced a piece of 20 cm twin-core PCF into single-mode fibers (SMF) on its two ends [17]. By modal coupling interference between two cores’ light, a sensitivity of 21 pm/MPa was achieved. Hu et al. developed a pressure sensor based on a side-hole two-core micro-structured optical fiber [18]. Different structures of fibers were manufactured and tested. The highest sensitivity was 110.8 pm/MPa on a fiber with 8 cm length and 200 µm diameter. However, the modal coupling principle required the sensing element to be at least several centimeters, which was not suitable for point pressure measurement. Moreover, a pressure sensor based on the Sagnac interferometer was created for downhole applications. A 60 cm polarization-maintaining photonic crystal fiber (PM-PCF) was coiled circularly to a diameter of 1.8 cm [19]. It was tested up to 20 MPa at 293 °C. Its highest sensitivity was 4.21 nm/MPa. However, its large size hampered its applications in restricted space situations.

Fabry–Perot interferometers are widely used for their simple structure, easy demodulation method and point measurement ability [20,21,22,23]. Wang et al. proposed an interferometric intensity-based optical fiber sensor [24]. A lead-in single-mode fiber and a reflecting single-mode fiber were thermally fused inside a hollow silica glass tube with an internal diameter of 127 µm with a certain air gap separation between the two fibers. The pressure sensitivity of this sensor was about 15.5 pm/MPa. However, the use of the hollow glass tube introduced a larger outer diameter. The interface of the SMF and the tube formed a protuberance, which could be a weak point of the sensor under pressure.

Wu et al. spliced a 2.1 mm solid-core PCF to an SMF, and the other end of the PCF was collapsed by arc discharging to increase the reflectivity [25]. The outside diameter of this structure was uniform, and was only 125 µm. Its sensitivity was tested to be 2.8 pm/MPa from 0 to 400 MPa, and it could survive at up to 700 °C. The in-line fiber pressure sensor was simple to construct, repeatable and easy to handle. Table 1 summarizes the above-mentioned sensors’ sensitivities, pressure ranges and structures.

By using a silica capillary tube, we built a new compact structure, composed of a section of capillary tube whose internal and outside diameter is 75/125 µm. Due to its hollow structure, it could have a higher sensitivity than the sensor in [25]. Moreover, its sensing element could be as short as several hundreds of micrometers, thus achieving point measurement. A section of the capillary tube was spliced with SMFs on its two ends. The two SMF/tube interfaces acted as an FP interferometer (FPI). Moreover, its pressure sensitivity was further increased by the Vernier effect.

The Vernier effect was originally employed by Vernier calipers to enhance the accuracy of length measurement. By taking advantage of the small-scale difference between the main ruler and the Vernier, the measurement sensitivity could be further increased by an order of magnitude. In recent years, the Vernier effect has also been applied in fiber optic sensing by using two interferometers as fixed and sliding parts of the Vernier scale [26,27]. Several configurations have been proposed to achieve the Vernier effect, such as two cascaded FPIs [23,28], two cascaded MZIs [29,30,31], two cascaded fiber loops [32], a cascaded Sagnac loop and an FP cavity [33,34]. All these configurations were cascaded configurations, which means that the sensors were fabricated along one fiber. This configuration led to a compact structure but increased the complexity of the structure, and it was difficult to keep one interferometer (as a fixed part of the Vernier-scale) insensitive and make the other (as a sliding part of the Vernier-scale) sensitive to the measured parameter. A parallel configuration has also been also studied by researchers [35,36,37]. More than one interferometer with the same structure and method but with different sensing lengths were fabricated and then connected in parallel to create a Vernier effect. A parallel-structured Vernier effect was used for our sensor, increasing its sensitivity by up to 6 times.

## 2. Methodology

### 2.1. Principle

An FP interferometer was built within a 125 µm diameter fiber (SMF-28, Corning Inc., Corning, NY, USA). A lead-in SMF and a reflecting SMF were spliced to a silica capillary tube (CAP075/150/24T, Fiberguide Industries, Caldwell, ID) with a certain air gap separation between the two SMFs, as shown in Figure 1.

The inner diameter *r_i_* of the tube was 75 µm, and the outer diameter *r_o_* was 125 µm, the same as that of the SMF. A beam of incident light was reflected successively by the two SMF/tube interfaces. Two reflected light beams inferenced with each other and generated an interference pattern. When ambient pressure increased, the structure would deform to change the distance between two interfaces, and the interference pattern would shift accordingly.

### 2.2. Fabrication

The splicing between the SMF and the tube was facilitated by a fusion splicer (S177A FITEL, Furukawa, Japan). The SMF and the tube were spliced together by arc discharging using a similar method as demonstrated in [38].

An SMF was positioned with a forward distance of about 50 µm between its end face and the electrode center, as shown in Figure 2a. Then, we manually aligned the SMF and the tube in the x and y axes, and adjusted the gap between their end faces to be about 10 µm. Finally, they were pressed together longitudinally by the mechanical force exerted from the splicer’s micro-positioning blocks and softened by an arc discharge. The tube’s inner cavity deformed partially after splicing, as shown in Figure 2b, but since the air/fiber interface still existed, the light beam would be reflected on the interface. The outer diameter slightly changed and formed a curve; thus, it would not hamper the whole structure’s strength under lateral pressure. This method is complicated and requires practice and experience. The fusion splicer we used does not allow us to control its arc intensity, but in practice, we found that as long as the forward distance and the tube/SMF end face gap are well maintained, the splicing results are acceptable, though not identical.

The tube part was then cut to a specific length by a fiber cleaver (CT-30, Fujikura, Japan). Then, it was spliced again with another section of cleaved SMF. The two SMF/tube (air) interfaces acted as two reflective surfaces of an FP interferometer. A micrograph of an obtained pressure sensor is shown in Figure 3. The length of the tube L was measured to be 430 µm by comparing it with the sensor’s outer diameter of 125 µm.

Two reflective beams interfered with each other and generated an interference pattern that could be detected by an optical sensing analyzer (OSA, si720, Micron Optics, Roanoke, VA, USA). The spectrum of this sensor is shown in Figure 4. The two adjacent peaks were at 1549.20 nm and 1551.99 nm. Based on Equation (1) [20], the length of this interferometer was calculated to be 430.88 µm, close to the measured value.
(1)$${L=\lambda }_{1}\lambda_{2}/2n\left(\lambda_{2}{-\lambda }_{1}\right)$$

## 3. Experiments and Results

### 3.1. Pressure Test

The sensor was tested inside stacks of ultra-high-molecular-weight polyethylene (UHMWPE) composite fabric. It was put in the center of a 2.5 × 2.5 inch fabric layer with adhesive used on A and B to fix the sensor, as shown in Figure 5c. Then, another five layers of fabric were placed on the sensor and another four layers were placed below. Thus, the sensor was inside 10 layers of fabric. The sample setup was prestressed overnight. The stacks were then compressed by the Instron 5500R machine to simulate the composite formation process.

Loadings were applied using a cylindrical fixture with a diameter of 2.25 inches, on a flat plate, as shown in Figure 5b. The loadings and the corresponding pressure values are shown in Table 2, where the pressure was calculated by the loading divided by the bearing area (2.25^2^π/4 squared inches). During the test, the sensor’s spectrum at each loading was recorded by OSA and saved in a computer in real time. The whole setup is shown in Figure 5a.

### 3.2. Results and Discussion

The sensor’s spectra from 1545 to 1550 nm at different loadings are shown in Figure 6. It showed a blue shift along the wavelength axis as the loading increased. It also showed an attenuation in intensity, which might be caused by the deformation of the tube and the bending of the lead-in fiber. We used the wavelength shift to represent the pressure change because the wavelength shift would not be affected by the fiber’s curving, bending or other fluctuation induced by environmental disturbances.

Tracking the shift values at different loadings, the relationship between the spectra shifts and the loading/pressure is shown in Figure 7. A linear fit was conducted. The sensor’s pressure sensitivity was measured to be 0.29 pm/psi (42.2 pm/MPa) with an R-square of 0.99. The bare FBG sensor’s pressure sensitivity is typically 3.04 pm/MPa [15]; our sensor’s sensitivity was 14 times higher than the bare FBG.

### 3.3. Temperature Test

This sensor’s temperature survivability and temperature cross-sensitivity were also essential. They were tested inside a temperature chamber (Imperial II Radiant Heat Oven, Lab-line, Melrose Park, IL, USA) from 20 °C to 140 °C with a step of 20 °C. Sensor spectra were recorded in three heating and cooling cycles. The wavelength shifts at different temperatures are shown in Figure 8.

The biggest error value of two data points at the same temperature was 0.005 nm. The spectra shift range in the whole experiment was 0.102 nm. The hysteresis was 0.005/0.102 = 4.9%, which is low, so the sensor’s temperature response repeatability was suitable. A linear fit shows the temperature sensitivity of 0.833 pm/°C with an adj. R-square of 0.99, as shown in Figure 9.

The sensor’s pressure sensitivity was 42.2 pm/MPa, and its temperature sensitivity was 0.833 pm/°C, so its temperature cross-sensitivity was 0.833/42.2 = 0.0197 MPa/°C.

Table 3 shows a comparison between the sensor with the bare FBG sensor regarding their performances. The FP lateral pressure sensor has a 14-fold higher pressure sensitivity than the bare FBG and has a temperature cross-sensitivity that is 200 times smaller.

## 4. Vernier Effect

We designed a parallel structured fiber optic FP sensor, as shown in Figure 10. The 430 µm sensor was used as Sensor 1. Sensor 2 was made using the same method as Sensor 1. Sensor 2’s cavity length L2 was close but not equal to that of L1. In this case, L2 = 365 µm. The two sensors were connected to the optical sensing analyzer through a 1 × 2 50:50 splitter.

Figure 11 shows the spectra of Sensor 1, Sensor 2 and their superimposed spectrum detected by the OSA. From Figure 11, we can derive the two sensors’ free spectrum ranges: *FSR*_1_ = 2.79 nm, *FSR*_2_ = 3.26 nm.

By choosing suitable cavity lengths to make the *FSR* of one sensor close but not equal to the other sensor, the Vernier effect could be achieved. Their superposition consisted of a series of fringes with different amplitudes. The *FSR* of the superposition spectrum envelope could be calculated as [26]:(2)$$FSR_{env}=FSR_{1}\times FSR_{2}/\left|FSR_{1}-FSR_{2}\right|$$

The theoretical *FSR* of the superposition spectrum envelope was 19.35 nm, consistent with the measured value from Figure 11, 19.75 nm.

Sensor 1 was tested inside 10 fabric layers on the Instron 5500R machine under loadings from 1250 to 8750 lb (pressure from 314 to 2200 psi), while Sensor 2 was left free of pressure. Sensor 1 acted as the sliding part of the Vernier scale, while Sensor 2 acted as the fixed part. The superposition spectra at different pressures are shown in Figure 12left. The red curves mark the envelopes of the spectra. We plotted the envelopes only in Figure 12right. While pressure increased, the Sensor 1 spectrum shifted, and the envelope shift was magnified by a certain factor [26] to be 5.94 theoretically:(3)$$M=FSR_{sliding}/\left|FSR_{sliding}-FSR_{fixed}\right|$$

Figure 13 shows the envelopes at different loadings. Figure 14 shows the shift values’ linear fit. The sensitivity was 1.75 pm/psi with an R-square of 0.99. Sensor 1’s pressure sensitivity was 0.29 pm/psi. The amplification was calculated to be 1.75/0.29 = 6.03, close to the theoretical value.

## 5. Summary and Discussion

A fiber lateral pressure sensor based on the Vernier effect-improved Fabry–Perot interferometer was fabricated with a uniform outer diameter of 125 µm. Due to the hollow structure, the sensor had a higher sensitivity than sensors fabricated from solid-core fiber. Its pressure sensitivity was 14 times higher than bare FBG fiber. Moreover, by using the Vernier effect, its sensitivity was further improved 6-fold. It could survive 2200 psi and 140 °C. Its small sensing length made it effective for point pressure measurement. It could work for real-time pressure monitoring at a certain point during composite-forming procedures.

Moreover, it can be further developed into a multi-point sensor by building multiple sensors together. The sensor’s free spectrum range is determined by the length of the glass tube. Sensors with different sensing lengths can be put at different points and then connected to the optical sensing analyzer through an optical splitter. The spectrum superposition’s frequency components will be obtained by Fast Fourier Transform (FFT). In the frequency domain, sensors with different lengths have different frequencies. Then, by using a specific filter and inverse FFT, the spectrum from a single sensor can be extracted. By tracking the spectrum shift of each sensor, the pressure at different locations can be detected.

The drawback of this method is that the fitted envelope has a gentler spectrum. Thus, the measurement of the envelope’s spectrum shift FSR is less accurate compared with the single FP’s original spectrum. In Table 4, we listed the exact spectrum shift values plotted in Figure 7 and Figure 14, and then calculated the magnified values and their standard deviation to provide an impression of the difference between two measurements.

## Figures and Tables

**Figure 1 sensors-22-07850-f001:**
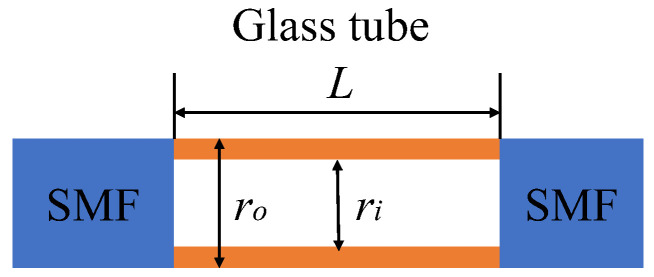
The design of the fiber lateral pressure sensor.

**Figure 2 sensors-22-07850-f002:**
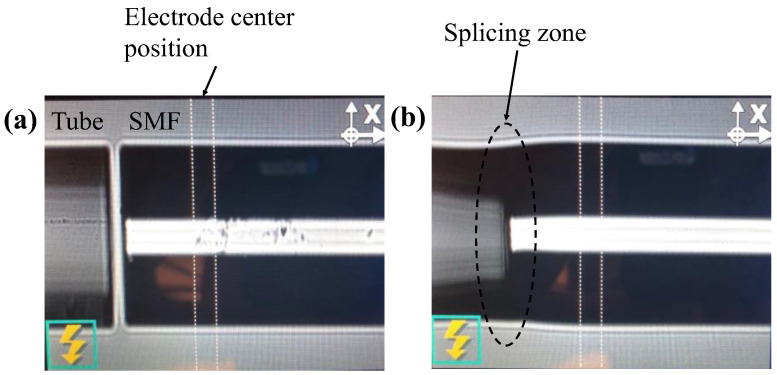
Images of the splice joints (**a**) before and (**b**) after splicing.

**Figure 3 sensors-22-07850-f003:**
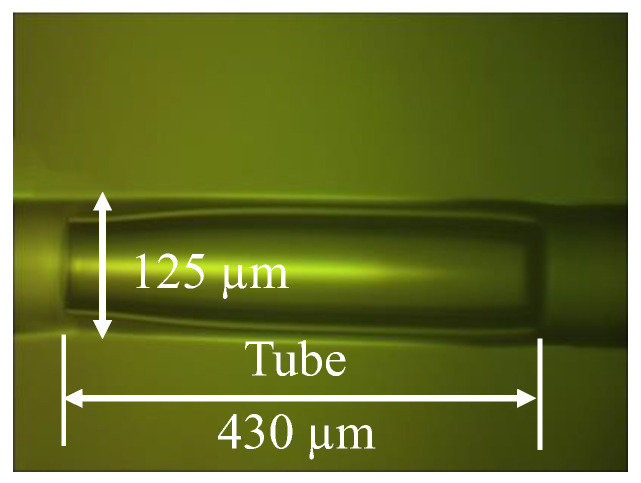
The micrograph of an obtained lateral pressure sensor.

**Figure 4 sensors-22-07850-f004:**
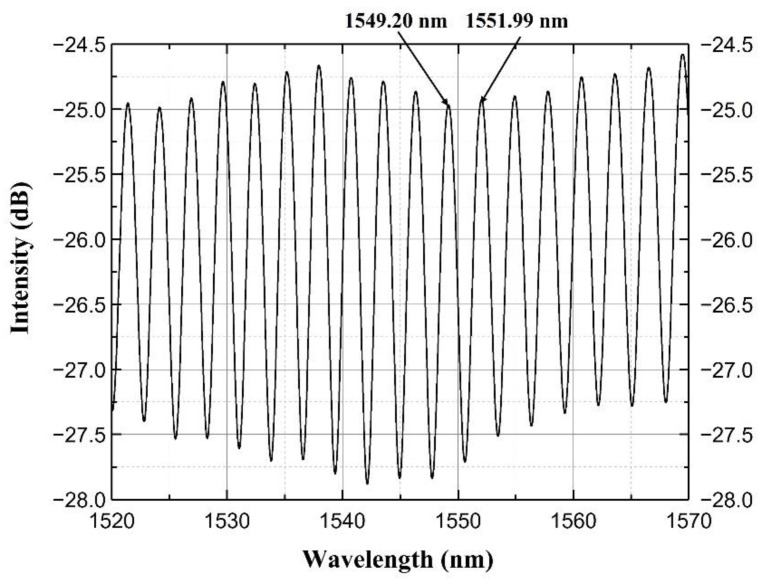
The spectrum of the lateral pressure sensor.

**Figure 5 sensors-22-07850-f005:**
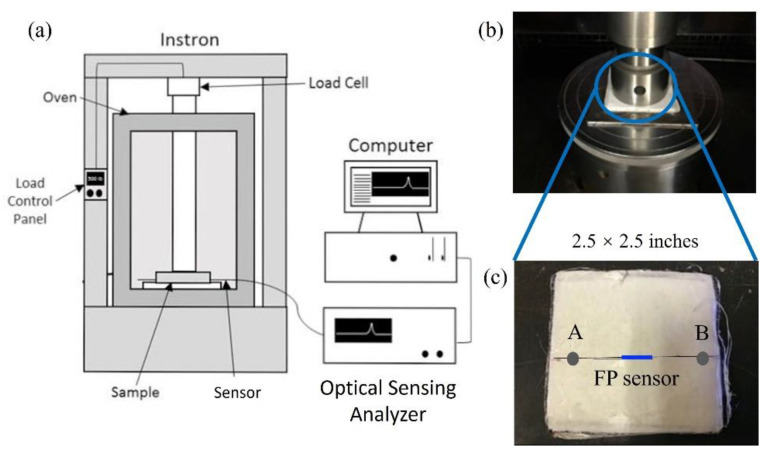
(**a**) The whole setup of the compression test. (**b**) The stacks under compression. (**c**) The sensor fixed on a layer of fabric.

**Figure 6 sensors-22-07850-f006:**
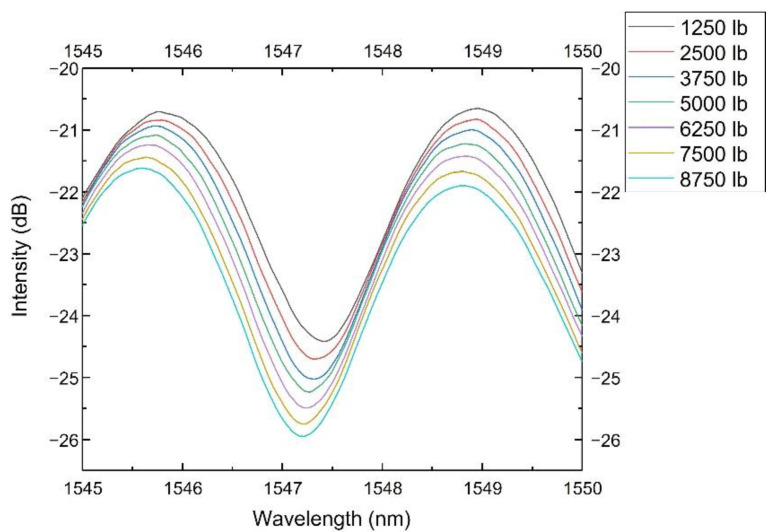
The sensor spectra at different loadings (from 1545 to 1550 nm).

**Figure 7 sensors-22-07850-f007:**
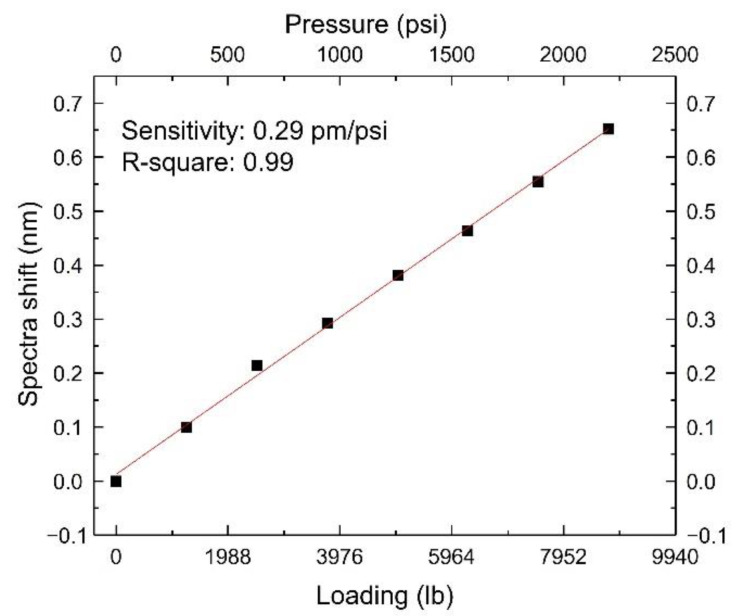
A linear fit between the spectra shifts and the loading/pressure.

**Figure 8 sensors-22-07850-f008:**
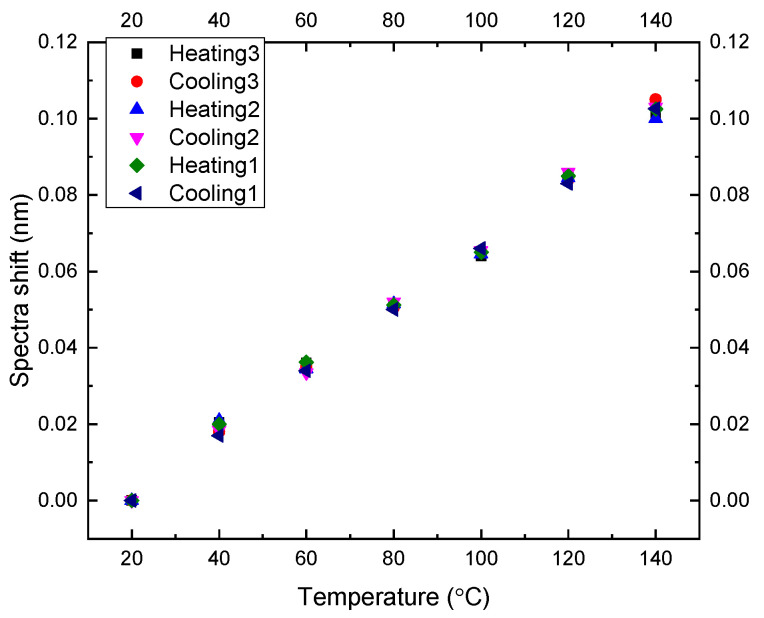
Spectra shifts in three heating and cooling cycles.

**Figure 9 sensors-22-07850-f009:**
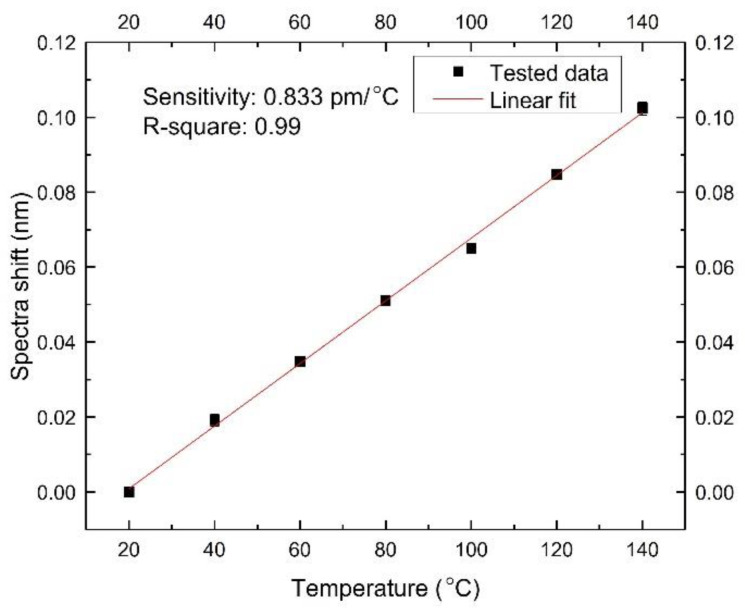
A linear fit between the spectra shifts and the temperature.

**Figure 10 sensors-22-07850-f010:**
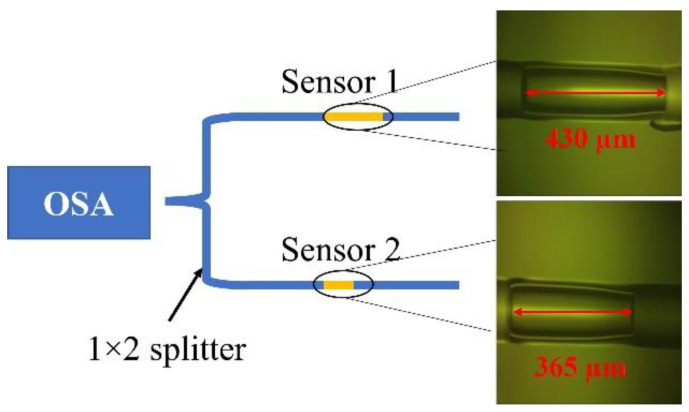
The schematics of the parallel FPIs pressure sensor.

**Figure 11 sensors-22-07850-f011:**
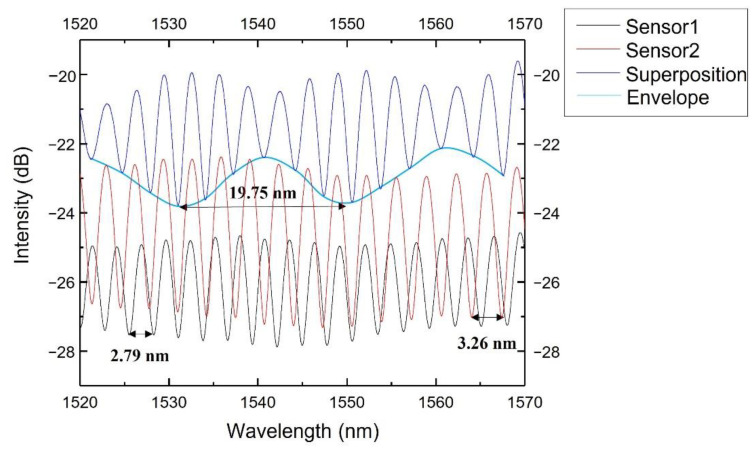
The spectrum of Sensor 1, Sensor 2 and their superposition.

**Figure 12 sensors-22-07850-f012:**
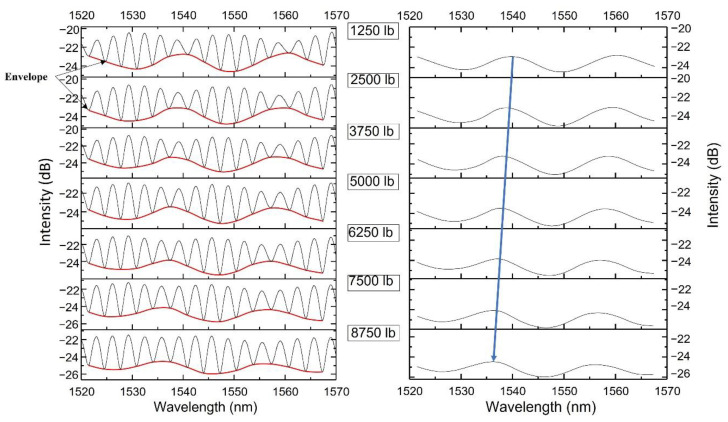
(**left**) Superposition spectrum at different pressures. (**right**) Envelope shift under pressure.

**Figure 13 sensors-22-07850-f013:**
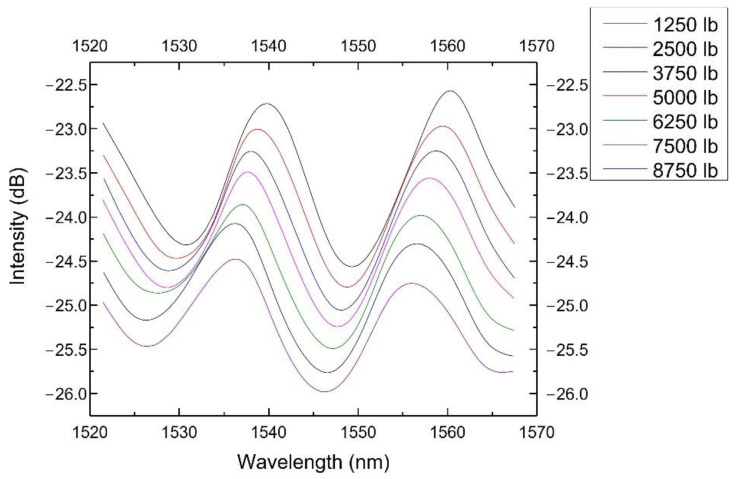
The envelopes of spectra at different loadings.

**Figure 14 sensors-22-07850-f014:**
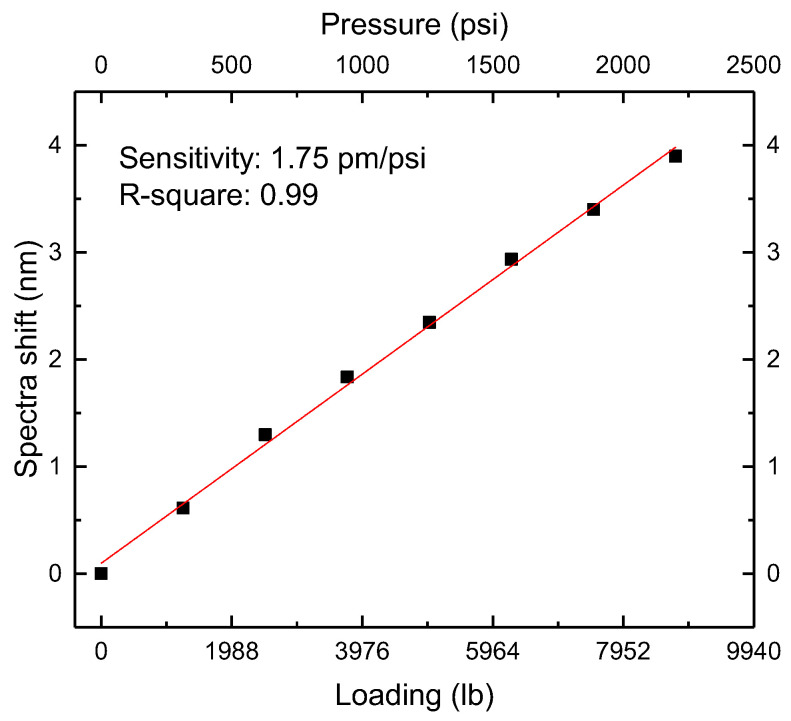
The linear fit of parallel FP sensors’ superposition envelope shifts.

**Table 1 sensors-22-07850-t001:** Summary of the fiber optic lateral pressure sensors.

Group	Sensitivity (pm/MPa)	Pressure Range (MPa)	Structure
C. Wu et al. [16]	12.86	25	FBG structure on a grapefruit micro-structured fiber
Z. Liu et al. [17]	21	45	A 20 cm twin-core photonic crystal fiber
G. Hu et al. [18]	110.8	200	An 8 cm side-hole two-core micro-structured optical fiber
H. Fu et al. [19]	4210	20	A 60 cm polarization-maintaining photonic crystal fiber was coiled circularly to a diameter of 1.8 cm
A. Wang et al. [24]	15.5	40	Two SMFs were fused inside a hollow silica glass tube
C. Wu et al. [25]	2.8	400	A 2.1 mm solid-core PCF to an SMF and the other end of the PCF was collapsed

**Table 2 sensors-22-07850-t002:** The applied loadings and the corresponding pressure values.

Loading (lb)	Pressure (psi)
1250	314.4
2500	628.8
3750	943.1
5000	1257.5
6250	1571.9
7500	1886.3
8750	2200.7

**Table 3 sensors-22-07850-t003:** The comparison between the FP sensor and the bare FBG sensor.

	FP Sensor	FBG Sensor
Pressure sensitivity (pm/MPa)	42.2	3.04
Temperature sensitivity (pm/°C)	0.833	12
Temperature cross-sensitivity (MPa/°C)	0.0197	3.95

**Table 4 sensors-22-07850-t004:** The spectrum shift values comparison between single and Vernier FP.

Loading (lb)	Single (nm)	FSR (nm)	M Value
1250	0.10	0.61	6.10
2500	0.21	1.30	6.04
3750	0.29	1.83	6.25
5000	0.38	2.35	6.16
6250	0.46	2.93	6.32
7500	0.55	3.40	6.13
8750	0.65	3.90	5.97
			Std: 0.12

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
