# Peer review of "Fiber Lateral Pressure Sensor Based on Vernier– Effect Improved Fabry–Perot Interferometer"

_sensors, 2022, doi:10.3390/s22207850_

Round 1

Reviewer 1 Report

In this work. authors fabricated a FP interferometer as pressure sensor. I don't find any significant scientific mistake, and the work is interesting. Some minor comments as below

1. Author should indicate the distance of air-gap, which is important for readers to counter check with the FSR.

2. Please include hysteresis test result in Figure 7.

3. Please check the unit used in Figure 10.

Author Response

Thank you for these valuable comments! Please see my address below.

  1. Author should indicate the distance of air-gap, which is important for readers to counter check with the FSR.

Response: In this pressure sensor, the distance of air-gap is the distance between two SMFs ends. The sensor used for part 3 has an air gap of 430 µm, shown in Figure 3; the sensors used for part 4 have a 430 µm and 365 µm air gap separately, shown in Figure 10.

  1. Please include hysteresis test result in Figure 7.

Response: There is no hysteresis for Figure 7, because the deformation of fabric layers during the test is not recoverable. So, with current setup it is unable to unload and then reload pressure. We are building a pressure chamber to apply pressure on the sensor through oil, to make the testing procedure repeatable.

  1. Please check the unit used in Figure 10.

Response: Done.

Reviewer 2 Report

 A fiber optic pressure sensor based on fused-spliced SMF-silica capillary tube-SMF sandwich structure is proposed and demonstrated in this study. The sensor can survive 2200 psi and 140 °C, and the obtained pressure sensitivity of 42.2 pm/MPa is 14 times higher than that of bare FBG. The sensitivity can be further improved by 6 times through Vernier effect. This is a meaningful study. But, I would suggest that authors should address the following issues clearly to make it more suitable for publication.

1. Some recent literatures on lateral pressure should be including in the paper, So that readers can have a better understanding of the latest progress in this field. Such as IEEE Photonics Technology Letters, 2021, 24 (22): 2038-2041. Optics & Laser Technology, 143 (2021) 107354. Journal of Lightwave Technology, 2022, 40 (12):3935-3941. IEEE Photonics Journal, 2022, 14(1): 6803808.

2. In Fabrication of the sensing structure, the splicing SMF to the tube by arc discharging is the difficulty. How reproducible and consistent is this method?

3. What is the effect of capillary length on the sensitivity of the sensor?

4. The graph format of the experimental results in this paper is not uniform, some have borders, while some do not. It is suggested to modify the unified.

5. There are some mistakes in the text, such as “125- m” in page 4, line 102, should be “125μm”. The sensor size mark in Figure 10 shows garbled characters.

Author Response

Thank the reviewer for these comments. We have revised the manuscript. Please see our detailed responses in the attached file. 

Reviewer 3 Report

Comments for the article

"Fiber Lateral Pressure Sensor Based on Vernier-effect Improved Fabry-Perot Interferometer"

in MDPI Sensors

A fiber optic pressure sensor that can survive 2200 psi and 140 °C based on the Fabry-Perot Interferometer was proposed. The method of increasing the sensor sensitivity by 6 times through the Vernier effect was described. Great work, actually, but there are several comments to the paper, they are listed below:

1)      Line 102, 125 μm, please, check units

2)      The main question about the whole work is about the used interrogation techniques. Were the spectral measurements by OSA performed in real-time continually or just in certain moments? Is it possible to use the proposed interrogation approach for real-time measurements? Please, discuss it.

3)      I would suggest investigating the Eq. (2) for optimal FSR values, which can provide the maximum sensitivity. Whether it’s effective to decrease FSR2 in comparison to FSR1 to provide better sensitivity or not?

The overall quality of the paper is quite good. I would recommend it for publication after the proposed corrections.

Author Response

Appreciate these comments! Please see our responses below. 

  • Line 102, 125 μm, please, check units.

Response: Done.

  • The main question about the whole work is about the used interrogation techniques. Were the spectral measurements by OSA performed in real-time continually or just in certain moments? Is it possible to use the proposed interrogation approach for real-time measurements? Please, discuss it.

Response: The measurements were real-time. The OSA sampling frequency is 5Hz. So, it is possible to use the proposed interrogation approach for real-time measurements.

  • I would suggest investigating the Eq. (2) for optimal FSR values, which can provide the maximum sensitivity. Whether it’s effective to decrease FSR2 in comparison to FSR1 to provide better sensitivity or not?

Response: Yes, it is effective to further decrease FSR2 (by increasing the length of Sensor2 to be closer to the length of FSR1) to provide higher sensitivity (In Eq3, smaller |FSRsliding – FSRfixed| leads to larger M.) There is no maximum sensitivity since M can be infinite theoretically. But it may bring other concerns. For example, a larger FSRenv could be exceed the monitoring range of OSA, making it harder to track the spectrum shift (like how many cycles have passed). To guarantee two complete cycles spectrum and some surplus, FSRenv of 19.75nm was applied here.